# Comparison of Mineral, Metabolic, and Oxidative Profile of Saanen Goat during Lactation with Different Mediterranean Breed Clusters under the Same Environmental Conditions

**DOI:** 10.3390/ani10030432

**Published:** 2020-03-04

**Authors:** Carmen L. Manuelian, Aristide Maggiolino, Massimo De Marchi, Salvatore Claps, Luigi Esposito, Domenico Rufrano, Elisabetta Casalino, Alessandra Tateo, Gianluca Neglia, Pasquale De Palo

**Affiliations:** 1Department of Agronomy, Food, Natural resources, Animals and Environment, University of Padova, Viale dell’Università 16, 35020 Legnaro, Italy; carmenloreto.manuelianfuste@unipd.it (C.L.M.); massimo.demarchi@unipd.it (M.D.M.); 2Department of Veterinary Medicine, University of Bari Aldo Moro, Casamassima km 3, 70010 Valenzano, Italy; elisabetta.casalino@uniba.it (E.C.); alessandra.tateo@uniba.it (A.T.); pasquale.depalo@uniba.it (P.D.P); 3Council for Agricultural Research and Agricultural Economy Analysis-Research Centre for Animal Production and Aquaculture, S.S.7 Via Appia, 85051 Bella Muro, Italy; salvatore.claps@crea.gov.it (S.C.);; 4Department of Veterinary Medicine and Animal Production (DMVPA), University of Naples Federico II, Naples, Via Federico Delpino 1, 80137 Napoli, Italy; luigespo@unina.it (L.E.); neglia@unina.it (G.N.)

**Keywords:** goats, Mediterranean heritage breed, blood chemistry, negative energy balance, blood mineral profile, oxidative stress

## Abstract

**Simple Summary:**

The study aims to compare physiological and productive responses of five Mediterranean heritage goat breeds with a high production breed, spread worldwide, the Saanen breed. The overall objective of the paper was to highlight that in difficult environmental conditions, although not so extreme, a high production breed does not actually represent the best solution, both from animals’ welfare conditions and for production. Mediterranean breeds showed better milk quality than Saanen, and tended to recover earlier from negative energy balance. However, no differences were observed on long-term oxidative stress markers. This paper contributes to increase interest towards animal biodiversity and valorization of local breeds, which are characterized by a long selective pressure linked to the environmental adaptation and less selected for productive traits.

**Abstract:**

This study aimed to describe metabolic, oxidative, and mineral blood profiles of Saanen does through lactation compared with Mediterranean breed clusters (Maltese and Rossa Mediterranea, and Jonica, Garganica, and Girgentana). Milk and blood samples of 57 dairy goats (9–10 goats per breed) were collected from the 2nd to the 30th week of lactation every 2–3 weeks. Saanen showed greater milk yield and somatic cell score, and lower fat and protein percentage through lactation (*p* < 0.05) than the Mediterranean breed clusters. Blood analysis revealed that stage of lactation had a greater impact than breed cluster, except for uric acid, alkaline phosphatase, and aspartate aminotransferase. Plasmatic non-esterified fatty acids indicated a greater negative energy balance in Saanen than the other breed clusters during early and medium lactation stages (*p* < 0.05). Serum Cl, Mg, and Ca increased in all the breed clusters from early to the following stages of lactation (*p* < 0.05). No significant prooxidant/antioxidant imbalances were detected in any of the three clusters during the entire lactation. In conclusion, Mediterranean breeds tended to recover earlier from negative energy balance than Saanen, but effects of breed or stage of lactation on long-term oxidative stress indicators were not evident.

## 1. Introduction

Animal genetic resources are the primary biological capital for livestock development and adaptability to altered conditions, such as climate change and consumer demands, and it is essential for sustainable rural development. However, studies have been mainly conducted in breeds that are spread worldwide because, for many years, the milk industry has focused on increasing milk production. Therefore, farmers have often replaced heritage breeds with worldwide spread ones to increase farm profitability. Thus, the characterization of local breeds in terms of physiology and production is crucial to make informed decisions on their management and strategies for their preservation. In 2017, Italy had the 5th largest population of goats (9.92 × 10^5^ heads [1]) in the European Union. According to DAD-IS from FAO [2], in Italy 59 different breeds are reared in Italy. Three breeds are classified as international, one breed as regional, and 55 breeds as local. From those local breeds, 60% are at risk of extinction, including the Girgentana (GI), Jonica (JO), and Rossa Mediterranea (RM); and 29% breeds are of unknown risk for extinction, including the Garganica (GA). In addition, GI, JO, RM, and GA are within the most important breeds reared in South Italy. A general description of the five breeds is available in Currò et al. [3], as well as milk yield, composition, and somatic cell score (SCS), milk mineral content [4], and milk fatty acids profile [4]. 

Plasma chemistry profiles associated to milk quality patterns provide useful information on nutritional status, metabolic condition, and on overall welfare status of does [5]. The Cl, Na, K, Mg, P, and Ca are considered macrominerals and blood plasma is used to assess the adequacy of mineral amounts in ruminants. Glucose, non-esterified fatty acids (NEFA), cholesterol, and triglycerides are indicative of animal’s energy status. Total protein, albumin, globulin, creatinine, acid uric, and blood urea nitrogen (BUN) evaluate animal’s protein status or renal function. Hepatic function is checked through alkaline phosphatase (ALT), aspartate aminotransferase (AST), alanine aminotransferase (ALP), and total bilirubin. Additionally, the oxidative status is assessed through thiobarbituric acid reactive substances (TBARS), lipid hydroperoxides (HYDROP), proteins carbonyls (PC), ferric reducing ability of plasma (FRAP), and superoxide dismutase (SOD). 

There is scarce information on the mineral, metabolic, and oxidative plasma profiles in goats, their evolution through lactation, and breed differences. In particular, breed differences of those parameters could reveal the adaptative capacity of heritage versus worldwide spread breeds to specific environmental conditions. Therefore, the evaluation of these parameters would reveal the capacity of adaptation of the Mediterranean goat breeds to the Mediterranean environment in comparison with animals selected from a productive point of view. Moreover, the results could be of interest in several other countries characterized by similar subtropical environmental/climatic conditions. The experimental hypothesis was that local breeds, although with lower production performances than other breeds, tend to be more suitable both from a welfare and milk quality point of view in subtropical environment conditions. Therefore, the aim of this study was to characterize the metabolic, oxidative, and mineral profile of Saanen (i.e., high yield) does through lactation compared with Mediterranean breeds that have medium (Maltese and Rossa Mediterranea breeds) and low (Jonica, Garganica, and Girgentana breeds) milk yield levels. 

## 2. Materials and Methods 

The study was carried out at the experimental farm of the Council for Agricultural Research and Economics (CREA, Potenza, Italy) from February to September 2018. Experimental procedures and animal care conditions followed the recommendations of European Union directive 86/609/EEC and were approved by the ethics committee (Ethical Code:PG/2019/0028161) for animal welfare from the University of Naples Federico II.

### 2.1. Animals

A description of the animals used, management, and feeding conditions is available at Currò et al. [3]. Briefly, a total of 57 dairy goats (9 or 10 does per breed) that kidded twins in February 2018 were enrolled in the study until the end of lactation (30 weeks). At the beginning of the study, does presented an average body condition score between 2.5 and 3.0 (1 = very thin to 5 = very fat, with 0.5 point-increment [6]) and an average breed body weight between 42 ± 6 (GI) and 64 ± 7 kg (SA). Parity ranged from 1 to 5 and was balanced among breeds. Milk and blood samples were collected from the 2nd to the 30th week of lactation, every two weeks, reaching a total of 855 samples for each fluid. As kids were kept with their dams until 40 days from birth, during that period kids were temporally separated from their dams 24 h before the sampling day.

### 2.2. Feeding and Pasture Quality

All the animals were allowed to graze in a natural pasture area of 1.78 ha, rectangular shaped, located in the South of Italy (40° 38’ N: 15° 49’ E) at 360 m above sea level. The pasture was divided into four equal paddocks with similar herbage availability and botanical composition. During the experimental period all the animals (regardless of the breed) alternately grazed in the four paddocks for 8 h/day. The duration of grazing in each paddock was modified according to the pasture quality, ranging from one to two weeks. The grass height reached the highest values during spring (45–58 cm) and the lowest ones during summer (7–16 cm). Samples for botanical composition assessment and for chemical analysis were taken monthly from February to October (nine sampling times) in all the paddocks where animals did not graze for at least one week. Ten squares of 1 m^2^ along one of the diagonals of each paddock were hand-cut at 4 cm stubble height. The herbage collected was used for the botanical assessment according to the dry weight rank method [7,8]. Thereafter, the samples were immediately oven-dried at 60 °C for 48 h and ground to pass a 1-mm screen using a Cyclotech mill (Tecator, Höganäs, Sweden). After mixing, the samples were analyzed for dry matter, crude protein, neutral detergent fiber, acid detergent fiber, acid detergent lignin, organic matter, and nutritive value, expressed as feed units for milk using the same techniques listed by Romanzin et al. [9]. Appendix A shows the botanical and chemical composition of the pasture, considering mean and standard deviation of each parameter among the nine sampling times. Botanical and nutritional composition of the pasture confirmed that the experimental conditions adopted during the study were overall representative of the Mediterranean grassland type [10,11].

Moreover, goats were supplemented in the shelter with hay (composition: 60%–65% of grasses and 35%–40% of legumes and others; chemical composition: 89.10% of dry matter and 15.10% of crude protein, 52.60% of neutral detergent fiber, and 1.10 Mcal/kg of net energy of lactation on dry matter basis) ad libitum; and supplemented in the milking parlor (morning and evening milkings) with commercial concentrate (Mangimi Losasso s.r.l., Baiano, Italy; chemical composition: 88.20% dry matter and 21.70% of crude protein, 23.00% of neutral detergent fiber, and 1.77 Mcal/kg of net energy of lactation on dry matter basis) according to their requirements considering the mean body weight and mean milk production for each breed following the National Research Council (NRC) (2007) recommendations for small ruminants, and adjusted every 15 days throughout lactation.

### 2.3. Environmental Climatic Conditions

Climatic data was recorded using a weather station recognized by the World Meteorological Organization [12] and furnished by the Italian Air Force. The weather station was placed 43 m far away from the northern border of the pasture area at the same altitude as the pasture. The environmental temperature and relative humidity were recorded every hour. This dataset was used to calculate the temperature humidity index (THI) values per hour according to the formula used by Bernabucci et al. [13]:THI = (1.8 × AT + 32) − (0.55 − 0.55 × RH) × [(1.8 × AT + 32) − 58](1)
where AT is the environmental temperature expressed in Celsius degrees—so that the term (1.8 × AT + 32) represents the conversion of temperature data in Fahrenheit degrees—and RH is the relative humidity. For each day, the maximum and minimum daily THI value was identified, and the daily average THI value was calculated as the arithmetic mean among the hourly THI values during the day. Data regarding the THI variation throughout the study is presented in Appendix A.

Although in the present study we did not record environmental climatic data in the stalls where does were kept after grazing, we consider the data reported for pasture as suitable for the stall environment because stalls were only 280 m away from the grazing area, and 315 m away from the weather station and at the same altitude. Moreover, does were kept in a 10 × 15 m open sided barn, provided with a flat concrete insulated roof of the same dimensions. During winter season, as well as along poor weather days (rainy and/or windy), from one to three sides of the barn were covered with semi-rigid plastic sheets with a maximum height of 2 m. These indoor conditions allow us to consider the climatic conditions found on the pasture directly correlated to those in the barn. Moreover, Bernabucci et al. [13] and Ravagnolo et al. [14] showed that the highest correlation with heat stress in cattle is the maximum daily THI, which occurs during daytime, when animals enrolled in the study were on pasture. 

Thus, we can conclude that environmental records retrieved during the experiment were typical of the Mediterranean area. They showed environmental temperatures during late spring and, above all, summer greater than the upper critical limit for goats proposed by Lu [15]. Therefore, during the study, the heat stress conditions in the experimental animals could be classified as moderate to severe for goat species [16,17,18].

### 2.4. Milk Sample Collection and Analysis

Milk sample collection and analysis has been described in Currò et al. [3]. Briefly, all goats were mechanically milked twice a day (0730 and 1730 h) in a double 24-stall parallel low-line milk pipeline milking parlor (Alfa Laval Agri; Monza, Italy) equipped with recording jars and electronic pulsators at a vacuum of 38 kPa, 90 pulses/min, and 60% pulsation ratio. Pre-milking included only forestripping, without any other preparation of udder and teats. Individual milk yield (kg/day) was recorded as the sum of morning and evening milkings using the recording jars in the milking parlor. Individual milk samples (50 mL) were collected during the morning milking, stored in portable refrigerators (4 °C) and transferred to the milk laboratory of the Breeders Association of Basilicata region (Potenza, Italy). Milk samples were warmed at 37 °C in a water bath prior to fat, protein, casein, and lactose analysis with MilkoScan FT6000 (Foss Electric, Hillerød, Denmark). Fat-corrected milk at 3.5% (FCM, kg/d) was calculated according to Pulina et al. [19]:FCM3.5% = milk yield (kg/day) × (0.634 + 0.1046 × fat %).(2)

Somatic cell count (SCC, cells/mL) was determined using Fossomatic FC (Foss Electric; Hillerød, Denmark) and transformed to somatic cell score (SCS) according to Wiggans and Shook [20]:SCS = 3 + log2 (SCC / 100,000).(3)

### 2.5. Blood Sample Collection and Analysis

Blood samples (18 mL) were collected by jugular venipuncture before the morning meal into vacuum tubes with a negative pressure system for serum (9-mL tubes without anticoagulant) and plasma (9-mL tubes with 15 USP U/mL of heparin; BD Vacutainer, Becton Dickinson, Canada Inc., Vacutainer 1, Oakville, Canada). All blood samples were collected the same day as the milk samples. All tubes were immediately centrifuged (plasma at 1500 × *g* for 10 min, 4 °C; serum at 2000 × *g* for 15 min, 4 °C). Blood serum and plasma were harvested and stored in aliquots (1 mL) at –20 °C for later analysis.

Clinical biochemistry parameters were obtained from the serum samples using an automated biochemistry analyzer (CS-300B; Dirui, Changchun, China) as described by De Palo et al. [21]. The following parameters were assessed: NEFA, cholesterol, triglycerides, glucose, plasma minerals [Cl, Na, K, inorganic P (iP), Ca], total proteins, albumin, uric acid, BUN, ALT, AST, ALP, and total bilirubin. Globulins and Albumin/Globulin ratio (Alb:Glob) were calculated starting from total protein and albumin parameters. Before beginning each analytical session, the standards furnished in the assay kits were used to calibrate the multi-parameter analyzer (Seracal, Gesan Production Kit, Campobello di Mazara, Trapani, Italy). After setting the calibration curve, two multi-parameter control sera (Seracontrol N and Seracontrol P, Gesan Production Kit, Campobello di Mazara, Trapani, Italy) were used to verify internal accuracy, considered satisfactory when the measured value deviated by no more than 3.00 % from the manufacturer’s declared values.

Plasma samples were used to calculate oxidation parameters: TBARS, HYDROP, PC, FRAP, and SOD. Thiobarbituric acid reactive substances were determined spectrophotometrically according to De Palo et al. [22], HYDROP were measured spectrophotometrically by an iodometric method, and PC levels were determined spectrophotometrically according to Reznick and Packer [23].

### 2.6. Statistical Analysis

Mediterranean breeds (GI, JO, RM, MA, and GA) were clustered according to their milk yield aptitude in the whole lactation [3] as medium production level breeds (MPLB: MA and JO) and low production level breeds (LPLB: GI, GA, and RM). The Saanen breed was considered separately from the other breeds as a high productive and Alpine origin breed. For the statistical analysis, lactation data were grouped in three stages as early lactation (EL: the first four weeks of lactation), medium lactation (ML: from 5th to 10th week of lactation), and late lactation (LL: from 11th to 30th week of lactation), according to results obtained by Maggiolino et al. [24].

For each variable, values that deviated more than three standard deviations (SD) from the respective mean were treated as missing values. First, data were analyzed using the MIXED procedure of SAS v. 9.4 (SAS Institute Inc., Cary, NC, USA) with repeated measures [25].The statistical analyses included breed cluster (SA, MPLB, and LPLB), stage of lactation (EL, ML, and LL), and their interaction as fixed effects, and animal effect nested within breed cluster and residual as random effects. As all the animals were at the same lactation stage during the same period, the environmental factors (climate and pasture quality) were overlapped to stage of lactation factor, so the study was not able to detect and discriminate the “environment” effect from the “lactation” one. Thus, although in the paper “stage of lactation” is reported as an independent effect, it represents both the effect of environmental conditions and of the mammary gland biology. Residuals for each variable after applying the statistical model were normally distributed, except for ALP that was Log10 transformed before applying the MIXED model. Estimated least square means (LSM) for the main effect were calculated and differences were evaluated using Tukey’s test when comparing the three stages of lactation within each breed cluster, and the Dunnett’s test when comparing the three breed clusters considering SA as the control group. Values are shown as least square mean (LSM) ± standard error (SE) and significance was declared at *p* < 0.05 unless otherwise indicated. 

## 3. Results

### 3.1. ANOVA and Descriptive Statistics

The ANOVA analysis revealed that milk production traits were all affected (*p <* 0.05) by breed cluster, except for protein and casein content, as well as breed affected plasma mineral and metabolic profile (Table 1). The breed clusters did not show effects on serum proteins profile and oxidative profile, with the exception of HYDROP (*p <* 0.05) and FRAP (*p <* 0.001). The effect of the stage of lactation was significant for all milk and blood parameters analyzed in the present study, except for K and ALP. The F-value of the model for breed cluster and stage of lactation effect indicated that stage of lactation usually had a greater (greater F-value) or similar impact (similar F-value) than breed cluster. Only uric acid, ALT, and AST revealed a greater breed cluster than stage of lactation effect.

The coefficient of variation (CV) in blood parameters was < 10% for Na (CV = 2.6%), Cl, total protein, albumin, and Mg (CV = 9.6%). The CV in the present study ranged from 10% to 20% for cholesterol and K (CV = 10.2%), Ca, globulin, glucose, PC, Alb:Glob, SOD, ALT, creatinine, HYDROP, and BUN (CV = 18.5%). Additionally, it was between 20% and 40% for AST (CV = 22.0%), triglycerides, iP, FRAP, TBARS, total bilirubin, acid uric, and NEFA (CV = 39.2%). The greatest variation in blood parameters was observed for ALP (CV = 97.1%).

### 3.2. Breed Cluster Effect within Lactation Stage

The effect of breed clusters on milk production is depicted in Table 2. During EL and ML stages of lactation, Saanen produced more milk and FCM than the two Mediterranean breed clusters (*p* < 0.05). However, during LL stage, Saanen only produced more milk (yield and FCM) than the LPLB cluster (*p* < 0.05). Saanen does yielded milk with lower fat content (*p* < 0.05) than the other clusters during the whole lactation, except for MPLB cluster during EL stage (*p* = 0.21). Milk protein and casein concentration during EL stage was lower in LPLB than in Saanen cluster (*p* < 0.05), but MPLB does did not differ from the Saanen cluster at this stage of lactation. In contrast, during ML stage, MPLB had a greater protein percentage than the Saanen cluster (*p* < 0.05), whereas both Mediterranean breed clusters showed a greater casein yield than Saanen (*p* < 0.05). These differences in milk protein and casein concentrations between Saanen and the Mediterranean breed clusters were not detected during LL stage. Although lactose concentrations in LPLB were greater than in the Saanen cluster for the whole lactation (*p* < 0.05), MPLB only revealed greater values than the Saanen cluster during EL and LL stages (*p* < 0.05). The SCS values recorded by Saanen breed were greater than those by both the Mediterranean breed clusters for the whole lactation (*p* < 0.05).

Table 3 shows the effect of breed clusters on mineral plasma profile at each stage of lactation. Chlorine and Na showed greater plasma concentration in LPLB than in the Saanen cluster during EL and ML stages (*p* < 0.05). Moreover, MPLB too had greater Na values than the Saanen cluster during EL stage (*p* < 0.05). On the contrary, K values were greater in Saanen than both Mediterranean breed clusters during ML and LL stages (*p* < 0.05); and MPLB had also lower concentrations of K than the Saanen cluster during EL stage (*p* < 0.05). Magnesium only differed between Saanen and MPLB clusters during LL stage (*p* < 0.05). While Ca concentrations were greater in LPLB and MPLB than the Saanen cluster during EL stage (*p* < 0.05), a lower Ca plasma concentration was recorded during LL stage in MPLB relative to the Saanen cluster (*p* = < 0.05). The iP values were lower during ML stage in both the Mediterranean breed clusters than in Saanen clusters (*p* < 0.05), and this difference was also maintained between MPLB and Saanen clusters during LL stage (*p* < 0.05).

Table 3 also reports the effect of breed clusters on plasma metabolite profiles at each stage of lactation. The NEFA concentration was greater in Saanen than in both the Mediterranean breed clusters during EL and ML stages (*p* < 0.05), and then, all three clusters reached almost the same values during LL stage. Plasma triglyceride concentrations were lower in the Saanen than in MPLB and LPLB clusters during EL stage (*p* < 0.05). Glucose levels did not differ during EL and ML stages, whereas greater values were present in MPLB than in the Saanen cluster during LL stage (*p* < 0.05). Creatinine was greater in LPLB than in the SA cluster during EL stage (*p* < 0.05), but Saanen values were greater than in MPLB and LPLB clusters during LL stage (*p* < 0.05). Uric acid was lower in both the Mediterranean breed clusters than in Saanen during EL stage (*p* < 0.05). Additionally, during ML and LL stages, the MPLB cluster maintained lower values of acid uric compared with the SA cluster (*p* < 0.05). At each stage of lactation, BUN did not differ among Saanen and the Mediterranean breed clusters, except for greater values in the LPLB than in the Saanen cluster during LL stage (*p* < 0.05). The LPLB cluster recorded greater ALT than Saanen during EL stage, whereas Saanen showed greater ALT than the MPLB cluster during LL stage. Saanen does had greater AST than the MPLB cluster during EL and ML stages (*p* < 0.05), whereas AST was greater in LPLB than in the Saanen cluster during LL stage (*p* < 0.05). The ALP was lower in Saanen than in the two Mediterranean breed clusters during both ML and LL stages (*p* < 0.05).

In general, the oxidative profile (Table 4) did not show many significant differences among Saanen and the other Mediterranean breed clusters within any stage of lactation, but a few exceptions were observed. Saanen presented greater TBARS values than the LPLB cluster during ML stage (*p* < 0.05); SOD activity was greater in MPLB than SA cluster during LL stage (*p* < 0.05); and, during LL stage, FRAP of MPLB and LPLB clusters were lower and greater than the Saanen cluster, respectively (*p* < 0.05).

### 3.3. Lactation Effect within Breed Clusters 

Table 5 shows the effect of lactation stage on milk production within each breed cluster. In all three breed clusters, milk yield was greater during EL and ML than during LL stage of lactation (*p* < 0.05). Although the same trend was observed in the Saanen cluster for FCM yield, MPLB and LPLB clusters showed greater FCM yield during ML than in both EL and LL stages (*p* < 0.05). Fat content in Saanen was lower during EL and ML than during LL stage (*p* < 0.05), whereas both the Mediterranean breed clusters presented an increasing trend in fat as lactation proceeds (*p* < 0.05). For protein percentage, the same trends reported for fat content was observed in Saanen and Mediterranean breed clusters (*p* < 0.05). In contrast, milk casein content did differ during lactation for Saanen breed, while an increasing trend as lactation proceeds was observed for the MPLB cluster as well as a lower casein concentration during EL versus ML and LL stages for the LPLB cluster (*p* < 0.05). Lactose content presented an opposite trend than the one described for fat content. That is, lactose content in Saanen was greater during EL and ML than during LL stages (*p* < 0.05), whereas both the Mediterranean breed clusters presented a decreasing trend as lactation advanced (*p* < 0.05). In all three breed clusters, the greatest SCS was recorded during LL stage (*p* < 0.05), although, for the LPLB cluster, SCS between LL and EL stages showed only a tendency to differ (*p* = 0.08).

Table 6 shows the effect of stage of lactation on the mineral concentrations within each breed cluster. Not all the mineral contents differed among lactation stage within each breed cluster. Serum Cl content in Saanen was lower during EL than during ML (*p* < 0.05) and during LL (*p* < 0.01). On the other hand, in the LPLB cluster, lower Cl concentrations were observed during LL than during EL (*p* < 0.05) and ML stages (*p* < 0.01). Variations of serum Na concentrations were only observed in the Mediterranean breed clusters with a decreasing tendency towards the end of lactation. Serum K and iP concentrations varied only in the MPLB cluster revealing greater content during EL than during ML and LL stages. In all three breed clusters, serum Mg and Ca content was lower during EL than during ML and LL stages, except for Ca content in LPLB between EL and ML stages (*p* = 0.40).

The effect of stage of lactation on plasma metabolite profiles within each breed cluster is also presented in Table 7. The NEFA profiles had a similar trend in both the Mediterranean clusters, with decreasing concentrations as lactation proceeds (*p* < 0.01). In contrast, Saanen showed lower NEFA only during LL compared with EL and ML stages (*p* < 0.01). Cholesterol varied only within the MPLB cluster, revealing greater values during EL and ML than LL stage (*p* < 0.01). Triglyceride differences across lactation stages were only detected in the Mediterranean breed clusters wherein concentrations decreased during LL relative to EL and ML stages (*p* < 0.01). Glucose values and BUN only differ in the Saanen and MPLB clusters, wherein both clusters had lower glucose concentrations during EL and ML than during LL stage (*p* < 0.01), and greater BUN concentration during EL and ML than during LL stage (*p* < 0.01). A lower creatinine content was observed in the MPLB and LPLB clusters during LL than during the previous lactation stages (*p* < 0.01). For uric acid, the greatest concentration was reached at ML stage in the MPLB cluster (*p* < 0.01), whereas in the LPLB cluster, ML and LL stages did not differ (*p* = 0.31). The Alb:Glob in MPLB and LPLB clusters was lower during EL than during ML and LL stages (*p* < 0.01). All three breed clusters revealed lower albumin concentration during EL than in the other lactation stages (*p* < 0.05), while globulins had greater values during EL stage of lactation only in MPLB (*p* < 0.01) and LPLB clusters (*p* < 0.05). In the Saanen cluster, ALT was greater during LL than during the previous lactation stages (LL vs. ML, *p* < 0.05; LL vs. EL, *p* < 0.01). In the LPLB cluster, AST reached the greatest values during LL compared with EL and ML stages (*p* < 0.01).

Table 7 displays the effect of stage of lactation on plasma oxidative parameters within each breed cluster. All three breed clusters showed greater TBARS, HYDROP, and SOD during LL than in previous lactation stages (*p* < 0.05). Plasma concentrations of PC and FRAP only differ in the LPLB and MPLB clusters. In LPLB does, PC were lower during LL versus EL (*p* < 0.01) and ML stages (*p* < 0.05). In MPLB, FRAP were also lower during LL than during EL and ML stages (*p* < 0.01).

## 4. Discussion

The paper tries to describe the variation in milk traits and some hematic parameters in goats according to their productive level (milk yield capacity) and lactation stage. The effect of stage of lactation in plasma minerals, metabolic, and oxidative parameters have previously been described in several dairy goat breeds [5,26,27,28], including studies with RM [29], also known as Red Syrian, and Saanen [30]. The present study mainly differs from those previously cited as ours aimed to compare heritage breeds clusters to Saanen breed, to help in the improvement of management and farming techniques in Northern Mediterranean areas and valorizing local breeds. 

Overall, all the values for serum electrolytes, chemical parameters, and enzymes were within the physiological ranges reported by Smith et al. [31] in goats, except for slightly greater values obtained for serum Cl, globulins, and total bilirubin, and greater AST. Average plasma Na and K levels were similar to those reported by Khaled et al. [32] in dairy goats during the first 3 months of lactation and by Krajničáková et al. [33] in white short-tailed primiparous goats considering the first 40 days postpartum. While Na is the main electrolyte responsible for maintaining volume and osmotic pressure of plasma, K is responsible for maintaining the intracellular osmotic pressure. Thus, the similarity between our results and the literature suggests that plasma Na and K levels are less influenced by extrinsic factors such stage of lactation. Our results also indicate that the blood concentration of Na and K is maintained within a narrower range than the levels for Ca, iP, and Mg. The fact that Ca, iP, and Mg concentrations differed from Khaled et al. [32] and Krajničáková et al. [33], even though Na and K concentrations were similar, could be related to the fact that Ca, iP, and Mg are primarily affected by the same homeostatic mechanisms.

The plasma NEFA and AST levels reported by Khaled et al. [32] in dairy goats and by Sadjadian et al. [30] in SA goats were in the range of the present study; however, they reported lower glucose (3.32 and 2.89 mmol/L, respectively) and BUN (8.87 and 8.08 mmol/L, respectively) levels with a similar total protein concentration (71.19 and 74.52 g/L, respectively). The AST concentration reported in the present study is greater than those reported by Omidi et al. [34] in Persian wild goats, and lower than the values reported by Perez et al. [35] in Spanish ibex. Many reasons may explain the AST high variability, considering that under the term “AST” we summarize several iso-enzymes found in liver, cardiac and skeletal muscles, kidneys, brain, lungs, pancreas, leucocytes, and erythrocytes [36]. A similar total protein level has also been observed in white short-tailed primiparous goats (67.30 g/L [33]) and in GI (72.26 g/L [37]). Relative to the present study, greater albumin levels have been observed in SA (35.65 g/L [30]) and GI goats (31.80 g/L [37]). Sadjadian et al. [30] also reported a lower cholesterol (2.74 mmol/L) and triglyceride (0.07 mmol/L) levels relative to this report. On the other hand, similar glucose (66.24 mg/dL) and total protein (6.96 mg/dL) levels but greater cholesterol levels (129.83 mg/dL) have been reported in Baladi goats [5]. 

There is still a lack of information regarding oxidative stress in ruminants, especially in goats [38]. Thiobarbituric acid reactive substances detect a range of lipid peroxidation products including malondialdehyde [38]. Serum TBARS have been reported to range from 1 to 12 µmol/L in Aardi goats during the first 4 weeks of lactation, which is greater than the range obtained in our study [39]. Moreover, the same authors have reported a SOD activity that ranged between 0.5 and 0.8 U/mL [39], which is below the range reported in the present study.

### 4.1. Milk Composition

The lower fat concentration in milk of SA compared to heritage breeds has already been reported by other authors [40,41]. That difference in fat percentage, as well as the differences recorded for other milk traits, could be, at least partially, explained by a dilution effect because SA is known to be a higher yielding breed than the heritage ones [3,42]. During early and mid-lactation, SA milk yield was higher than other breeds. Other studies have reported the same pattern for protein and casein concentrations in milk as the one described for fat when comparing SA with other heritage breeds [43,44]. However, we observed that SA milk protein and casein content was not highly different compared with the other Mediterranean breeds. In fact, the only differences found where during the early and medium lactation (i.e., the first third of lactation). The lower lactose content and greater SCS in SA relative to both Mediterranean breed clusters is consistent with other studies that reported that the goat breeds that show the greatest lactose concentrations are those with the lowest SCS, probably due to the decrease in mammary cell functionality as mastitis occurrence increases [45,46]. However, the negative correlation between lactose content and SCS has not been confirmed by other authors [47]. These contrasting results could be related to the physiological peculiarities of somatic cells in goat milk. In fact, SCS in goat species is not considered a good predictor of udder health status, due to the apocrine process of milk protein secretion which results in the release of the apical part of epithelial cells. In addition, the high variability of SCS is influenced by non-pathological factors (i.e., breed, parity, stage of lactation, type of birth, estrus, and diurnal, monthly, and seasonal variations) that are responsible for 48% of SCC variance in dairy goats [47]. Moreover, in the present study we recorded an increase of SCS during lactation in all three breed clusters, consistent with previous studies [48]. 

### 4.2. Blood Mineral Profile

In Baladi goats, Azab and Abdel-Maksoud [26] characterized an increase in plasma Ca, iP, Mg, Na, and K levels through the first month of lactation. Nubian dairy goats in early lactation presented lower plasma Ca and Mg and a greater Na levels than in mid-lactation depending on parity [27], in agreement with the results from the present study. A decrease in plasma Na from parturition until 28 days postpartum was also reported by Krajničáková et al. [33] in White short-tailed primiparous goats. However, those authors did not observe a decrease or increase of K, Ca, iP, and Mg plasma levels, probably due to the short period of their study which ended at 40 days postpartum. The lower Ca and Mg levels in early lactation could be related to the greater secretion of those two elements into milk due to the greater milk yield at the beginning of the lactation. Discrepancies with Ahmed et al. [27] results could be attributable to experimental design because they collected the milk samples from different goats for each stage of lactation. 

Scarce information is available in the literature regarding breed differences for plasma mineral profile in goats. Among minerals, the lower Na and Ca values during EL stage in Saanen does could be due to a higher mammary gland uptake for the higher milk yield. The differences of the Saanen relative to the Mediterranean breed clusters, during EL and ML, with regard to plasma Na (osmotic pressure of plasma) and K (intracellular osmotic pressure) agrees with the biological function of those two elements and with a greater galactopoietic rate, which results in the higher milk yield. 

### 4.3. Blood Metabolic Profile

The NEFA values reported in the present study are greater than those of some other studies [49,50], although overall similar to those reported by Hussain et al. [51]. As NEFA are strictly related to the energy balance of animals, and a NEFA concentration greater than 0.20–0.21 mmol/L indicates a catabolic condition [52,53], we can infer that all the breeds were in conditions of negative energy balance during the whole lactation, although the energetic deficit showed different intensities among the three groups. These results are consistent with the pasture quality and with the environmental conditions. Therefore, it appears that the study conditions did not allow animals to have enough energy supply from feeding and grazing to meet the energy requirements necessary for basal metabolism, milk production, grazing activities, and thermoregulation. In addition, during the same period, SA showed greater NEFA levels than the Mediterranean breed clusters. Further, as lactation advanced, in Mediterranean breed clusters there was a constant decrease of NEFA with a significant NEFA decrease between EL and ML stages, whereas in Saanen does the NEFA values in EL and ML stages remained severely elevated. This indicates not only a greater negative metabolic energy balance in Saanen versus the Mediterranean breeds at the same lactation stage, but also a delay in recovery of the energy balance relative to MPLB and LPLB clusters. Although the NEFA values indicated these energy balance conclusions, we did not find significant differences in serum glucose levels among the three breed clusters at each stage of lactation. Although glucose concentration is related to the metabolic status of the animals, and some papers reported a decrease of glucose during negative energy balance [49], our results could be considered consistent with the metabolic pathways activated during negative energy balance. Blood glucose level is regulated by efficient homeostatic systems, so a higher glucose consumption rate induces greater gluconeogenesis in liver and mobilization of body fat. This, in turn, increases NEFA as an alternative energy source for metabolism, maintaining constant blood glucose levels [51]. This may explain why NEFA values were affected by breed clusters but not glucose levels. Furthermore, only in Saanen and MPLB, the breeds with greater milk yield, revealed lower glucose levels during EL and ML than in LL stages. This observation is consistent with the role of glucose as precursor to lactose synthesis during lactation, although not a predictor of differences in negative energy balance intensity between the breeds. 

Triglycerides were lower in Saanen than in the two Mediterranean breed clusters during EL stage, confirming the negative correlation of this parameter with NEFA values, as an increase of triglycerides is ascribed to a re-esterification process, and thus a reduction of NEFA [49]. Moreover, triglycerides had different trends during lactation for the three breeds clusters. In the Saanen breed, we did not observe any significant variation along lactation, whereas in the Mediterranean breed clusters we recorded a decrease of triglycerides during LL stage, which is not consistent with the physiological indication of this parameter with energy balance. The present results disagree with Di Trana et al. [49], but it is possible that this decrease is related to an enhanced capacity of the mammary gland to uptake triglycerides to synthesize milk fat during LL stage [30,54]. Indeed, this hypothesis is consistent with the greater fat yield during LL stage in Mediterranean breed clusters than in SA.

Results observed for creatinine, uric acid, and BUN did not indicate any pathological conditions of renal activity. Overall, in all breed clusters, there is a consistent trend between creatinine and BUN levels, with a reduction of both these parameters during LL. This suggests that BUN changes could be due to an enhanced renal clearance, although this parameter is also strongly affected by forestomach recycling efficiency, protein intake, synthesis, and degradation [29,55]. Thus, our results are not able to fully explain the variation because we had no data to verify the absolute amount and quality of protein intake among the different breeds during lactation, and it is known that there is a breed effect on diet selection on pasture.

Serum protein profiles did not show substantial differences according to breed clusters, but revealed lower levels of albumin during EL than in the other stages of lactation, with an increase of Alb:Glob, mainly in Mediterranean breeds in ML and LL stages. Albumin acts as a carrier for lipids, hormones, vitamins, and minerals, it is involved in the immune function and cell regulation [56,57,58]. Additionally, as the main extracellular source of thiols, albumin is an important non-enzymatic antioxidant [29]. The reduction of albumin concentration during early lactation could be due to a compromised liver function related to the metabolic stress of the transition period and of the negative energy balance. 

The AST, ALT, and ALP enzyme values observed in the present study did not appear suitable as indicators of differences in metabolism among SA and Mediterranean breed clusters during lactation. Specifically, the high number of isoenzymes for each enzyme, and the wide diffusion of these enzymes in different tissues and cells, were associated with a rather high variability which may have precluded discrimination of meaningful trends in activity among breeds [59,60].

### 4.4. Oxidative Plasma Profile

Plasma oxidative status is the result of the interaction of many different compounds and systemic metabolic interactions [61]. In the present study, the trait that gives an overall evaluation of the oxidative balance of plasma is FRAP, a method to assess total antioxidant capacity in plasma [62]. Our results indicate that there were no significant prooxidant/antioxidant imbalances among breed clusters or at any stage of lactation. Although some metabolic stressors during the transition period and EL stage, such as heat stress, have been widely demonstrated to affect the oxidative status in ruminants [38], our study focuses on the whole lactation and not to shorter duration effects. This could explain our results in comparison with studies conducted for shorter periods and with higher sampling frequency across time. This hypothesis is consistent with the evaluation of results recorded on the concentration of oxidative metabolites (TBARS, HYDROP, and PC) in relation to antioxidant activity (SOD). Particularly, PC, which are the terminal products of proteins exposed to free radicals, did not show changes during our study. In contrast, indicators of lipid oxidation (TBARS and HYDROP) exhibited consistent changes for the three breed clusters throughout lactation, with an increase during LL stage. This greater concentration of end products from radical induced polyunsaturated fatty acids decomposition is negatively correlated to the SOD activity. The SOD represents the first defense against antioxidants among enzymatic antioxidants, and its activity showed greater levels in EL stage, in agreement with other reports [38]. As in the present paper we did not evaluate all the prooxidant and antioxidant systems, we are not able to fully explain the increase of lipid oxidation during LL stage, nor the pathway responsible of maintaining oxidative balance in plasma during the same stage of lactation. However, although TBARS assay has been widely considered inaccurate for the detection of MDA and HYDROP in animal fluids [21,38], we believe that these can be considered consistent results because of the application of the same technique for sampling and laboratory analysis, and because in the present study we compared between relative changes of groups across time and not absolute values. 

## 5. Conclusions

As expected, milk yield and composition were affected by breed cluster and lactation stage. In late lactation goats tend to produce less milk and less constituents, but more concentrate with higher percentage values. However, our study allowed us to characterize and highlight metabolic and oxidative parameters during lactation in different goat breed clusters: a highly productive Alpine breed and two clusters of Mediterranean breeds, differing for the milk production potential. Results showed that, despite the pasture quality and environmental climatic conditions, Mediterranean breeds tend to recover earlier than Saanen from the negative energy balance, which is consistent with their lower milk yield. Regarding oxidative status, the high variability of contrasting reactive oxygen metabolites among goat species, and the focus on the whole lactation, allow us to conclude that there are no effects of breed clusters or stage of lactation on long-term oxidative stress. These results might help in the management and conservation of the biodiversity in these species, such as increasing knowledge on the metabolic adaptation of heritage and worldwide spread breeds, with different milk production levels in Mediterranean farming conditions.

## Figures and Tables

**Table 1 animals-10-00432-t001:** Overall descriptive statistics of milk yield, somatic cell score, milk chemical composition, and metabolic, mineral, and oxidative plasmatic profile of goats in a complete lactation. It also reported the F-values and significance of the fixed effects included in the statistical model.

		Descriptive Statistics	Fixed Effect (F-Value)
Trait	N	Mean	SD	Minimum	Maximum	CV, %	Breed Cluster (BC)	Stage of Lactation (SL)	Interaction BC×SL


Milk									
MY, kg/day	778	1.17	0.52	0.10	2.85	44.5	74.76 ***	157.81 ***	7.14 ***
FCM, kg/day	777	1.20	0.49	0.12	2.73	40.7	48.35 ***	106.96 ***	5.50 ***
Fat, %	774	3.94	1.03	1.70	7.40	26.1	47.45 ***	125.32 ***	4.18 ***
Protein, %	771	3.33	0.54	2.10	5.37	16.2	2.45 ^†^	116.09 ***	2.34 *
Casein, %	772	2.45	0.40	1.57	3.94	16.3	1.90	75.47 ***	1.93 ^†^
Lactose, %	775	4.44	0.33	3.19	5.41	7.4	30.61 ***	199.31 ***	3.37 ***
SCS	784	5.57	2.00	-3.64	10.17	35.9	26.55 ***	55.72 ***	1.23
Serum Electrolytes									
Cl, mEq/L	776	111.89	3.14	97.50	123.30	2.8	4.43 ***	7.40 ***	4.18 ***
Na, mEq/L	776	146.08	3.79	135.60	158.20	2.6	7.54 ***	13.48 ***	2.13 *
K, mEq/L	781	4.23	0.43	3.01	6.43	10.2	42.96 ***	0.49	1.78
Mg, mg/dL	784	3.23	0.31	2.40	4.00	9.6	3.11 **	49.59***	1.47
iP, mg/dL	784	5.94	1.38	3.90	10.20	23.2	10.83 ***	7.48 ***	0.95
Ca, mg/dL	784	10.04	1.03	7.00	13.00	10.3	25.80 ***	82.33 ***	7.54 ***
Chemical parameters
NEFA, mmol/L	781	0.51	0.20	0.11	1.04	39.2	79.01 ***	424.94 ***	29.45 ***
Cholesterol mg/dL	782	90.93	9.31	75.60	134.60	10.2	4.57 **	10.84 ***	1.15
Triglycerides, mg/dL	783	34.18	7.71	15.40	56.20	22.6	7.31 ***	60.28 ***	4.45 ***
Glucose, mg/dL	783	63.12	7.84	47.50	79.00	12.4	2.91 *	6.76 ***	4.78 ***
Creatinine, mg/dL	777	0.82	0.14	0.50	1.30	17.1	3.59 **	31.99 ***	4.25 ***
Acid uric, mg/dL	769	0.46	0.17	0.10	1.00	37.0	20.01 ***	13.80 ***	4.70 ***
BUN, mg/dL	783	21.09	3.91	11.30	30.50	18.5	5.36 ***	31.18 ***	2.15 ***
Total bilirubin, mg/dL	773	0.20	0.07	0.10	0.40	35.0	11.33 ***	14.21 ***	8.27 ***
Serum Proteins
Total protein, g/dL	780	7.12	0.28	6.30	7.80	3.9	1.91	2.51 *	2.37 ^†^
Albumin, g/dL	779	2.89	0.20	2.40	3.50	6.9	1.62	25.02 ***	1.37
Globulin, g/dL	784	4.26	0.47	30	5.00	11.0	1.31	7.29 ***	2.52 *
Alb:Glob	779	0.69	0.10	0.50	1.13	14.5	1.77	15.50 ***	2.17 *
Serum Enzymes
ALT, IU/L	782	10.75	1.75	5.80	15.80	16.3	22.71 ***	5.99 ***	7.90 ***
AST, IU/L	783	226.08	49.76	109.40	481.10	22.0	118.09 ***	1.45 ***	12.64 ***
ALP, Log10 IU/L	764	2.17	0.23	1.91	3.27	10.551	11.74 ***	0.76	1.16
Oxidative status
TBARS, nmol/mL	779	0.88	0.29	0.12	1.79	33.0	0.93	186.27 ***	2.06 ^†^
HYDROP, µmol/mL	782	6.31	1.10	3.10	9.61	17.4	3.07 *	33.89 ***	1.84
PC, µmol/mL	783	110.14	13.84	24.54	168.45	12.6	0.94	21.82 ***	1.25 ***
FRAP, nmol/L	770	62.36	15.07	27.33	113.77	24.2	9.39 ***	26.26 ***	8.26 ***
SOD, IU/mL	751	107.92	17.39	50.61	183.45	16.1	0.17	65.55 ***	1.93 ^†^

MY= milk yield; FCM, = fat corrected milk at 3.5%; SCS = somatic cell score; iP = inorganic phosphorus; BUN = blood urea nitrogen Alb:Glob = Albumin:Globulin; ALT = alanine aminotransferase; AST = aspartate aminotransferase; ALP = alkaline phosphatase; TBARS = thiobarbituric acid reactive substances; HYDROP = lipid hydroperoxides; PC = protein carbonyls; FRAP = ferric reducing ability of plasma; SOD = superoxide dismutase; CV = coefficient of variation. † *p* < 0.10; * *p* < 0.05; ** *p* < 0.01; *** *p* < 0.001.

**Table 2 animals-10-00432-t002:** Milk production traits (least squares means and mean standard error) of the Saanen breed compared with Mediterranean medium production level breeds (MPLB: Maltese and Jonica breeds) and low production level breeds (LPLB: Rossa Mediterranea, Girgentana, and Garganica breeds) during early lactation (2–4 weeks in milking), medium lactation (5–10 weeks in milking), and late lactation (11–30 weeks in milking).

	Early Lactation				Medium Lactation				Late Lactation			
Trait	Saanen	MPLB	LPLB	MSE	Saanen	MPLB	LPLB	MSE	Saanen	MPLB	LPLB	MSE
MY, kg/day	1.784	1.395 *	1.144 *	0.25	1.997	1.369 *	1.213 *	0.10	1.026	0.907	0.73 *	0.11
FCM, kg/day	1.694	1.351 *	1.128 *	0.22	1.869	1.399 *	1.262 *	0.108	1.046	1.010	0.83 *	0.09
Fat, %	3.04	3.25	3.45 *	0.06	2.91	3.78 *	3.96 *	0.06	3.72	4.68 *	4.83 *	0.07
Protein, %	3.09	2.95	2.85 *	0.16	3.17	3.48 *	3.30	0.12	3.71	3.65	3.56	0.13
Casein, %	2.29	2.22	2.14 *	0.09	2.31	2.46 *	2.43 *	0.07	2.65	2.67	2.60	0.08
Lactose, %	4.51	4.71 *	4.73 *	0.06	4.46	4.48	4.59 *	0.04	4.02	4.19 *	4.33 *	0.08
SCS	5.95	4.86 *	4.91 *	2.53	5.76	4.89 *	4.41 *	2.7	7.79	6.28 *	5.91 *	3.10

MY = milk yield; FCM = fat corrected milk at 3.5%; SCS = somatic cell score; MSE = mean standard error. * The mean value significantly (*p* < 0.05) differs from the Saanen value at the same lactation stage.

**Table 3 animals-10-00432-t003:** Plasma mineral and metabolic profile (least squares means and mean standard error) of Saanen breed compared with Mediterranean medium production level breeds (MPLB: Maltese and Jonica breeds) and low production level breeds (LPLB: Rossa Mediterranea, Girgentana, and Garganica breeds) during early lactation (2–4 weeks in milking), medium lactation (5–10 weeks in milking), and late lactation (11–30 weeks in milking).

	Early Lactation				Medium Lactation				Late Lactation			
Trait	Saanen	MPLB	LPLB	MSE	Saanen	MPLB	LPLB	MSE	Saanen	MPLB	LPLB	MSE
Serum Electrolytes												
Cl, mEq/L	110.32	111.31	112.19 *	11.49	144.72	146.02	146.60 *	10.14	112.52	111.91	111.85	7.81
Na, mEq/L	145.34	147.84 *	147.82 *	13.86	144.72	146.01	146.59 *	13.55	145.07	145.22	145.21	12.74
K, mEq/L	4.44	4.19 *	4.35	0.22	4.58	4.11 *	4.21 *	0.15	4.55	4.10 *	4.22 *	0.12
Mg, mg/dL	3.01	3.08	3.11	0.10	3.34	3.32	3.37	0.08	3.32	3.23 *	3.32	0.08
iP, mg/dL	6.64	6.28	6.14	2.01	6.76	5.71 *	5.76 *	1.55	6.18	5.61 *	5.76	1.86
Ca, mg/dL	9.03	9.43 *	9.88 *	0.64	10.21	9.83	10.15	0.98	10.72	10.08 *	10.91	0.84
Chemical parameters												
NEFA, mmol/L	0.84	0.71 *	0.63 *	0.02	0.79	0.47 *	0.52 *	0.01	0.38	0.36	0.38	0.01
Cholesterol mg/dL	90.51	93.52	94.53	12.64	88.86	92.47	91.44	9.34	87.81	88.21	90.34 *	4.54
Triglycerides, mg/dL	33.57	38.85 *	37.81 *	3.49	36.55	37.59	34.15	4.27	31.00	30.58	29.45	5.78
Glucose, mg/dL	61.93	61.29	63.05	5.92	61.35	62.26	61.06	5.71	62.50	66.89 *	62.76	5.78
Creatinine, mg/dL	0.80	0.84	0.88 *	0.02	0.84	0.87	0.89	0.03	0.80	0.75 *	0.76 *	0.01
Acid uric, mg/dL	0.51	0.41 *	0.42 *	0.02	0.56	0.49 *	0.53	0.03	0.49	0.38 *	0.51	0.03
BUN, mg/dL	22.33	22.28	22.44	5.78	21.00	21.86	22.30	3.17	18.99	19.14	21.04 *	1.24
Total bilirubin, mg/dL	0.16	0.19 *	0.18	0.01	0.18	0.25 *	0.20	0.01	0.22	0.20 *	0.19 *	0.01
Serum Proteins												
Total protein, g/dL	7.05	7.13	7.10	0.11	7.20	7.08 *	7.12	0.07	7.21	7.10 *	7.16	0.06
Albumin, g/dL	2.81	2.83	2.76	0.03	2.92	2.94	2.95	0.04	2.90	2.91	2.89	0.04
Globulin, g/dL	4.31	4.34	4.42	0.27	4.28	4.18	4.06 *	0.18	4.37	4.21 *	4.29	0.20
Alb:Glob	0.66	0.66	0.63	0.001	0.69	0.71	0.72	0.001	0.67	0.70	0.68	0.001
Serum Enzymes												
ALT, IU/L	10.19	10.47	11.13 *	3.23	10.94	10.56	10.98	1.70	11.65	10.06 *	11.69	2.49
AST, IU/L	246.90	198.64 *	236.59	95.85	239.44	203.85 *	253.58	81.33	213.83	210.01	278.90 *	91.65
ALP, IU/L	148.76	200.27	171.28	42.97	132.61	187.07 *	192.93 *	26.79	121.75	192.06 *	187.80 *	26.74

MSE = mean standard error; iP = inorganic phosphorus; BUN = blood urea nitrogen, Alb:Glob = Albumin:Globulin; ALT = alanine aminotransferase; AST = aspartate aminotransferase; ALP = alkaline phosphatase, it was log 10-transformed to run the MIXED procedure of SAS, and then the least squares means were back-transformed to report the results. * The mean value significantly (*p* < 0.05) differs from the Saanen value at the same lactation stage.

**Table 4 animals-10-00432-t004:** Plasma oxidative profile (least squares means and mean standard error) of Saanen breed compared with Mediterranean medium production level breeds (MPLB: Maltese and Jonica breeds) and low production level breeds (LPLB: Rossa Mediterranea, Girgentana, and Garganica breeds) during early lactation (2–4 weeks in milking), medium lactation (5–10 weeks in milking), and late lactation (11–30 weeks in milking).

	Early Lactation				Medium Lactation				Late Lactation			
Trait	Saanen	MPLB	LPLB	MSE	Saanen	MPLB	LPLB	MSE	Saanen	MPLB	LPLB	MSE
TBARS, nmol/mL	0.87	0.83	0.81	0.01	0.71	0.69	0.63 *	0.02	1.00	1.08	1.07	0.05
HYDROP, µmol/mL	6.83	6.42	6.24	1.76	5.79	5.95	5.61	1.21	6.57	6.58	6.57	0.57
PC, µmol/mL	116.52	115.36	115.89	19.87	109.33	105.88	110.29	15.17	108.24	108.31	107.04	14.95
FRAP, nmol/L	63.69	68.47	69.19	32.47	62.67	65.70	65.91	26.60	57.58	51.84 *	64.58 *	18.19
SOD, IU/mL	120.65	118.35	122.03	12.48	105.66	104.17	102.17	12.72	101.41	105.55 *	101.59	13.07

MSE = mean standard error; TBARS = thiobarbituric acid reactive substances; HYDROP = lipid hydroperoxides; PC = protein carbonyls; FRAP = ferric reducing ability of plasma; SOD = superoxide dismutase. * The mean value significantly (*p* < 0.05) differs from the Saanen value at the same lactation stage.

**Table 5 animals-10-00432-t005:** Effect of the stage of lactation (early lactation, EL, 2–4 weeks in milking; medium lactation, ML, 5–10 weeks in milking; and late lactation, LL, 11–30 weeks) on the milk production traits of Saanen breed, Mediterranean medium production level breeds (MPLB: Maltese and Jonica breeds), and low production level breeds (LPLB: Rossa Mediterranea, Girgentana, and Garganica breeds).

	Saanen				MPLB				LPLB			
Trait	EL	ML	LL	MSE	EL	ML	LL	MSE	EL	ML	LL	MSE
Milk yield, kg/day	1.76 ^a^	1.97 ^a^	1.31 ^b^	0.09	1.28 ^a^	1.38 ^a^	1.15 ^b^	0.04	1.12 ^a^	1.21 ^a^	0.92 ^b^	0.04
FCM, kg/day	1.67 ^a^	1.86 ^a^	1.32 ^b^	0.09	1.25 ^a^	1.42 ^b^	1.27 ^a^	0.04	1.09 ^a^	1.25 ^b^	1.04 ^a^	0.03
Fat, %	2.97 ^a^	2.92 ^a^	3.56 ^b^	0.12	3.30 ^a^	3.85 ^b^	4.56 ^c^	0.07	3.42 ^a^	3.97 ^b^	4.70 ^c^	0.12
Protein, %	3.08 ^a^	3.16 ^a^	3.47 ^b^	0.09	2.94 ^a^	3.35 ^b^	3.55 ^c^	0.04	2.87 ^a^	3.29 ^b^	3.45 ^c^	0.05
Casein, %	2.28	2.30	2.48	0.07	2.21 ^a^	2.46 ^b^	2.59 ^c^	0.07	2.13 ^a^	2.43 ^b^	2.51 ^b^	0.03
Lactose, %	4.48 ^a^	4.47a	4.15 ^b^	0.05	4.67 ^a^	4.48 ^b^	4.29 ^c^	0.02	4.73 ^a^	4.59 ^b^	4.39 ^c^	0.03
SCS	5.98 ^a^	5.91 ^a^	7.77 ^b^	0.32	4.72 ^a^	4.95 ^a^	5.92 ^b^	0.17	4.80 ^a^	4.33 ^a^	5.49 ^b^	0.22

MSE = mean standard error; FCM = fat corrected milk at 3.5%; SCS = somatic cell score. a, b, c Different letters in the same line for each breed cluster indicate a significant difference (*p* < 0.05) between means.

**Table 6 animals-10-00432-t006:** Effect of the stage of lactation (early lactation, EL, 2–4 weeks in milking; medium lactation, ML, 5–10 weeks in milking; and late lactation, LL, 11–30 weeks) on the plasma mineral and metabolic profile of Saanen breed, Mediterranean medium production level breeds (MPLB: Maltese and Jonica breeds), and low production level breeds (LPLB: Rossa Mediterranea, Girgentana, and Garganica breeds).

	Saanen				MPLB				LPLB			
Trait	EL	ML	LL	MSE	EL	ML	LL	MSE	EL	ML	LL	MSE
*Serum Electrolytes*												
Cl, mEq/L	111.14 ^a^	112.99 ^b^	113.52 ^b^	0.51	111.76	111.63	111.80	0.34	112.62 ^a^	113.06 ^a^	111.29 ^b^	0.35
Na, mEq/L	144.84	144.86	145.34	0.70	147.95 ^a^	146.13 ^b^	145.37 ^b^	0.45	147.85 ^a^	146.89	145.47 ^b^	0.52
K, mEq/L	4.53	4.60	4.54	0.08	4.28 ^a^	4.12 ^b^	4.13 ^b^	0.04	4.35	4.22	4.20	0.05
Mg, mg/dL	3.00 ^a^	3.35 ^b^	3.30 ^b^	0.05	3.10 ^a^	3.31 ^b^	3.20 ^c^	0.03	3.08 ^a^	3.34 ^b^	3.38 ^b^	0.04
iP, mg/dL	6.70	6.75	6.58	0.26	6.33 ^a^	5.69 ^b^	5.83 ^c^	0.13	6.22	5.85	6.07	0.18
Ca, mg/dL	8.95 ^a^	10.08 ^b^	10.87 ^c^	0.20	9.46 ^a^	9.85 ^b^	10.19 ^c^	0.08	9.95 ^a^	10.20 ^a^	11.16 ^b^	0.13
*Chemical parameters*												
NEFA, mmol/L	0.85 ^a^	0.80 ^a^	0.38 ^b^	0.02	0.73 ^a^	0.47 ^b^	0.33 ^c^	0.01	0.65 ^a^	0.52 ^b^	0.35 ^c^	0.02
Cholesterol mg/dL	89.15	89.52	87.76	1.78	93.43 ^a^	92.40 ^a^	88.40 ^b^	0.92	93.70	91.10	92.87	1.34
Triglycerides, mg/dL	33.55	36.56	32.30	1.28	38.35 ^a^	37.30 ^a^	31.66 ^b^	0.72	37.18 ^a^	34.38 ^a^	30.06 ^b^	0.94
Glucose, mg/dL	59.75 ^a^	61.41 ^a^	66.90 ^b^	1.60	60.10 ^a^	61.71 ^a^	67.07 ^b^	1.42	62.55	60.66	62.84	1.00
Creatinine, mg/dL	0.80	0.85	0.79	0.02	0.84 ^a^	0.87 ^a^	0.72 ^b^	0.01	0.90 ^a^	0.88 ^a^	0.72 ^b^	0.02
Acid uric, mg/dL	0.49	0.56	0.58	0.04	0.40 ^a^	0.48 ^b^	0.39 ^a^	0.01	0.42 ^a^	0.53b	0.58b	0.02
BUN, mg/dL	22.78 ^a^	20.75 ^a^	18.21 ^b^	0.75	21.96 ^a^	22.19 ^a^	19.31 ^b^	0.39	22.28	21.96	21.02	0.48
Total bilirubin, mg/dL	0.14 ^a^	0.17 ^a^	0.22 ^b^	0.01	0.19 ^a^	0.24 ^b^	0.19 ^a^	0.01	0.16 ^a^	0.20 ^b^	0.18	0.01
*Serum Proteins*												
Total protein, g/dL	7.0 ^a^	7.23 ^b^	7.18	0.07	7.18	7.0	7.1	0.03	7.08	7.12	7.15	0.04
Albumin, g/dL	2.79 ^a^	2.93 ^b^	2.93 ^b^	0.04	2.83 ^a^	2.93 ^b^	2.92 ^b^	0.02	2.75 ^a^	2.96 ^b^	2.90 ^b^	0.03
Globulin, g/dL	4.23	4.32	4.26	0.76	4.42 ^a^	4.19 ^b^	4.23 ^b^	0.39	4.38 ^a^	4.15 ^b^	4.24	0.48
Alb:Glob	0.67	0.69	0.70	0.02	0.65 ^a^	0.70 ^b^	0.70 ^b^	0.01	0.64 ^a^	0.72 ^b^	0.69 ^b^	0.01
*Serum Enzymes*												
ALT, IU/L	10.46 ^a^	10.89 ^a^	12.07 ^b^	0.30	10.71	10.57	10.22	0.17	11.08	11.03	11.75	0.23
AST, IU/L	247.77	230.34	212.67	14.41	199.11	203.07	208.11	2.90	237.92 ^a^	252.20 ^a^	286.94 ^b^	6.83
ALP, IU/L	134.92	132.57	137.16	11.20	213.25	179.45	187.38	21.85	169.08	179.50	193.38	13.04

MSE = mean standard error; iP = inorganic phosphorus; BUN = blood urea nitrogen; Alb:Glob = Albumin:Globulin; ALT = alanine aminotransferase; AST = aspartate aminotransferase; ALP = alkaline phosphatase, it was log 10-transformed to run the MIXED procedure of SAS, and then the least squares means were back-transformed to report the results. a, b, c Different letters in the same line for each breed cluster indicate a significant difference (*p* < 0.05) between means.

**Table 7 animals-10-00432-t007:** Effect of the stage of lactation (early lactation, EL, 2–4 weeks in milking; medium lactation, ML, 5–10 weeks in milking; and late lactation, LL, 11–30 weeks) on the plasma oxidative profile of Saanen breed, Mediterranean medium production level breeds (MPLB: Maltese and Jonica breeds), and low production level breeds (LPLB: Rossa Mediterranea, Girgentana, and Garganica breeds).

	Saanen				MPLB				LPLB			
	EL	ML	LL	MSE	EL	ML	LL	MSE	EL	ML	LL	MSE
TBARS, nmol/mL	0.73 ^a^	0.72 ^a^	0.98 ^b^	0.03	0.71 ^a^	0.69 ^a^	1.03 ^b^	0.02	0.77 ^a^	0.64 ^b^	1.07 ^c^	0.03
HYDROP, µmol/mL	6.37	5.85 ^a^	6.51 ^b^	0.17	6.17 ^a^	5.95 ^a^	6.59 ^b^	0.12	6.07 ^a^	5.62 ^a^	6.55 ^b^	0.14
PC µmol/mL	111.26	109.64	107.61	2.78	108.92	105.74	108.21	1.34	113.04 ^a^	111.20 ^a^	106.19 ^b^	1.44
FRAP, nmol/L	60.08	61.87	63.53	2.98	65.95 ^a^	65.48 ^a^	50.59 ^b^	1.39	69.23	66.86	65.91	1.89
SOD, IU/mL	116.95 ^a^	105.94 ^b^	102.05 ^b^	7.78	115.77 ^a^	104.02 ^b^	106.42 ^b^	1.74	118.78 ^a^	102.37 ^b^	101.42 ^b^	2.13

MSE = mean standard error; TBARS = thiobarbituric acid reactive substances; HYDROP = lipid hydroperoxides; PC = protein carbonyls; FRAP = ferric reducing ability of plasma; SOD = superoxide dismutase. a, b, c Different letters in the same line for each breed cluster indicate a significant difference (*p* < 0.05) between means.

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
