# Peer review of "Comparison of Mineral, Metabolic, and Oxidative Profile of Saanen Goat during Lactation with Different Mediterranean Breed Clusters under the Same Environmental Conditions"

_animals, 2020, doi:10.3390/ani10030432_

Round 1
Reviewer 1 Report
na
Reviewer 2 Report
Title breed rather than breeds.
Simple summary: alter sentence on environmental conditions to ' in moderately challenging environmental conditions'.
Alter not differences to no differences.
Introduction
Change 'worldwide spread' to 'world dominating'
line 59 missing reference
Results
General comment for tables/table titles. Are minimum and maximum values required. Stick to SD or MSE . At present there is a mixture. Change least squares means to mean
Discussion
line 396 Variation in milk traits
Red text in conclusion - clister needs changing to cluster
This manuscript is a resubmission of an earlier submission. The following is a list of the peer review reports and author responses from that submission.
Round 1
Reviewer 1 Report
REVIEW: Animals
For this study authors compare plasma metabolites across three groups of goats: Saanen, which represents high milk producer, most selected world-wide for milk production; and 2 groups of medium production and low production autochthonous Mediterranean does during early, mid and late lactation.
In general this is a well written manuscript with a lot of data....which makes it very long. Which gets to me biggest criticism, and that is how can you get people/attract people to read this. It’s data driven, without hypothesis exploration—which may be important to studies that aim at underlying SNP, etc., driving phenotypic, adaptive differences, and understanding strengths of different breeds for sustainable farming in environmentally stressful situations….but really doesn’t add to our understanding of anything.
A few suggestions:
Shorten and summarize Move some data to supplementary sections Tell us a bit more about breeds so we care….scientists, what can this add. This will move this paper from a niche characterization of a few breeds, to using genomics, etc., to understand phenotype and adpatations. Nothing is told about size or phenotype or adaptive phenotypes…I think this stuff is interesting…selection can help with the understanding of evolution, how females adapt to stress of lactation, etc etc.Other comments:
The simple summary statement needs to be written/edited (by a native English speaker)
Suggest defining autochthonous—or use term heritage breed (that’s what we do in the US)—as I am not sure this is correct use of the word; certainly cosmopolitan is not typically used—maybe ‘most’ common domesticated breed of dairy goats
Authors use ‘genotype’ throughout, have the animals been genotyped? do we know they are different?..probably yes..but sometimes phenotype arises without mutation.
I think in results discuss milk production first, and then mineral, etc., as this is the ‘phenotype’ that authors divide groups by…but really never describe the animals (which is most interesting)
typically animals in a greater negative energy balance have a higher fat to protein yield (i.e. fatter milk), the difference in fat content of milk between Sanaan and others would suggest breed difference and not necessarily energy balance differences. The higher NEFA in Sanaan in EL and ML relative to other breeds suggest, they may be in a greater NEB state
Author Response
Dear Editor and Reviewers, first of all we need to express our gratitude for the effort in suggesting and evaluating the paper, clearly aiming to improve the quality of our draft.
As you will see, we accepted overall all the suggestions and opinions, except for few ones, where the issue was mainly a lack of clarity of the paper more than an error. So, in this case we improved clarity, trying to avoid the misunderstanding that generated doubts and suggestions by reviewers (and that potentially could generate doubts and misunderstanding by future readers).
Below we pasted the report of both the reviewers, reporting analytically in red our rebuttal or answers.
For this study authors compare plasma metabolites across three groups of goats: Saanen, which represents high milk producer, most selected world-wide for milk production; and 2 groups of medium production and low production autochthonous Mediterranean does during early, mid and late lactation.
In general this is a well written manuscript with a lot of data.... which makes it very long. Which gets to me biggest criticism, and that is how can you get people/attract people to read this. It’s data driven, without hypothesis exploration—which may be important to studies that aim at underlying SNP, etc., driving phenotypic, adaptive differences, and understanding strengths of different breeds for sustainable farming in environmentally stressful situations….but really doesn’t add to our understanding of anything.
Au: We appreciate your opinion on the paper. We agree on the high amount of data reported, so we decided to add on “Supplementary materials” all the data regarding environmental conditions, trying to reduce the length of the paper and focusing exclusively on the results obtained. However, we could try to reduce the paper length by deleting results on Milk production traits and focusing exclusively on blood parameters, if the reviewer considers it suitable for improving the paper. Although we explore here clusters of breeds, information regarding the same animals and breeds is available in the AAB paper (reference 3 of the manuscript).
Probably the aim of the paper in the original versions is not well highlighted, so we improved this part in the final part of the Introduction section.
A few suggestions:
Shorten and summarize Move some data to supplementary sections
Au: We moved data regarding environmental characterization to supplementary files. Moreover, we refer to other papers (lines 61 - 63) for breeds information.
Tell us a bit more about breeds so we care….scientists, what can this add. This will move this paper from a niche characterization of a few breeds, to using genomics, etc., to understand phenotype and adpatations. Nothing is told about size or phenotype or adaptive phenotypes…I think this stuff is interesting…selection can help with the understanding of evolution, how females adapt to stress of lactation, etc etc.
Au: We integrated this information in the Introduction section. A general description of the breeds included in the present study have been previously published in Reference 3. Also, milk yield, composition and SCS of the animals and breeds included in the present study is available in Reference 3, milk mineral content in Reference 4, and milk fatty acid composition in Reference 4. (Lines 61 - 63).
Other comments:
The simple summary statement needs to be written/edited (by a native English speaker)
Au: We deeply modified simple summary and it was edited by a native English speaker
Suggest defining autochthonous—or use term heritage breed (that’s what we do in the US)—as I am not sure this is correct use of the word; certainly cosmopolitan is not typically used—maybe ‘most’ common domesticated breed of dairy goats
Au: We changed “autochthonous breeds” with “heritage breeds” and “cosmopolitan” with “worldwide spread breeds”
Authors use ‘genotype’ throughout, have the animals been genotyped? do we know they are different?..probably yes..but sometimes phenotype arises without mutation.
Au: Although all the considered breeds have been already evaluated with genomic techniques, confirming that all these groups, for number of heads and for genomic features, can be scientifically considered “genotyped breeds”, we preferred to change “genotype” with “breed”, that is a term less directly linked to genomic definition, although it includes it also.
I think in results discuss milk production first, and then mineral, etc., as this is the ‘phenotype’ that authors divide groups by…but really never describe the animals (which is most interesting)
Au: We changed the order of the results according the suggestion. Breeds are better described in the Introduction section and the references where milk yield, composition, SCS, milk mineral content and milk FA profile have been published. We prefer not to directly include that information in the present paper, because it is already available in open access.
typically animals in a greater negative energy balance have a higher fat to protein yield (i.e. fatter milk), the difference in fat content of milk between Sanaan and others would suggest breed difference and not necessarily energy balance differences. The higher NEFA in Sanaan in EL and ML relative to other breeds suggest, they may be in a greater NEB state
Au: We added the hypothesis of higher basal NEFA levels in Mediterranean heritage breeds in the discussion section, although we did not approached the “fat to protein” yield, as we did not find clear supporting literature on the modification of this parameter in goat species, while we read several papers dealing with cows.
Reviewer 2 Report
Title needs more clarity with regards to ‘genotypes clusters’Mineral not mentioned as a result in the Abstract or in key search terms. Key terms need a complete re-think & goat not mentioned
Abstract purely descriptive with no key statistics
Parameters of interest mentioned but not expanded upon in terms of why they were selected for investigation in the introduction. The introduction as it stands is weak. Also lacking on information under standardised conditions on the digestive efficiency and feed conversion efficiency of the breeds of goat selected for the study. Suggested type of missing reference….(see N. Salah, D. Sauvant and H. Archimède (2014) Nutritional requirements of sheep, goats and cattle in warm climates: a meta-analysis. Animal, 8:9, pp 1439–1447 doi:10.1017/S1751731114001153)
Why were these particular breeds selected for the study – an indication of their relative influence/importance in the caprine dairy industry in the EU would be useful.
Not enough emphasis on the influence of dry conditions on parameters of interest in the Introduction and the fact that the literature for plasma and milk parameters of nutritional status health production ais biased towards temperate breeds and environmental conditions. More emphasis on potential effects of climate change – could this term go in the title?
Why measure environmental conditions and not incorporate them in your statistical model? Either as raw data or as some recognised index of thermal load? (see Ingram and Mount 2012 Man and Animals in hot environments)
Are all your parameters normally distributed? Some of the values for coefficient of variation are very high – did you check for normality in your data? Would a generalized linear model that can cope with mixed distributions of data: data with a bimodal or with a Poisson distribution be better?
Line 48 Plasma chemistry profiles… what about parallel milk chemistry profiles . A vague sentence. There needs to be more clarification on why selected parameters were investigated. The statements about the usefulness of these plasma chemistry profiles are unreferenced.
Line 82 I don’t understand the sampling procedure for nutritional content of the pasture. This would have varied considerably over the 30 week lactation cycle and yet the 9 sampling events were pooled to give one mean and variation value for each pasture variable over the course of the study. Why were pasture sampling events not tied to blood and milk sampling events? The data in Table 1 need to be broken down to each of the 9 sampling events
Lines 98-104 would be better tabulated
Table 1 and Fig 1 belong in the results section
Re-phrase lines 143-149
Do the addresses of the manufacturers of equipment belong in the text?
Lines 150-167 would be clearer tabulated
Lines 224-229 you should comment on lactation stage and lactation stage x genotype effects
Lines 321-242 Table 3 commentary. What strikes me is the lack of breed effects in the data? Would an alternative mode of statistical testing be worthwhile as you have been working with parameters with a high degree if inherent variation and possibly not a normal distribution – and have applied very stringent statistical testing for normal data. The other way forward if your data are not normal is to try transforming it but I have rarely had much luck with this approach. The most appropriate statistical test would be a better approach
Lines 251-278 – this is a blow by blow account of each result in Table 3. A briefer more condensed analysis of the main findings would be more appropriate.
Lines 274-278 You have achieved this with your description of findings from Table 4.
Table 6, Table 7 & Table 8 Superscript the letters indicating level of statistical significance
Lines 329-351 again too long winded – see previous comment about describing main findings from a Table of results
Discussion
Start discussion with the experimental aims
Lines 388 – 431 of Discussion; ANOVA and Descriptive Statistics; generally a descriptive comparison with data from the literature with insufficient inference – the following section is similarly unfocussed. Start from the perspective of the main themes in the experimental design: Mediterranean conditions; breed; lactation stage; (thermal load from the environment). Much of the data in literature you are making comparisons with has not been collected under comparable conditions and this is an important point to highlight within the framework of climate change.
Line 394 What environmental conditions were these data gathered under? This comment applied to any citation referencing clinical data.
Lines 404-406 Na and K levels can both be influenced by other factors such as hydration status; thermal load? So I don’t understand this statement ?
General comment for Discussion: You are not focussing on the environmental impact on the parmeters of interest. . This could also be incorporated as an index of thermal load into your statistical model. Environmental impact (or influence of thermal load) on the data is a novel and relevant theme that has been generally neglected in the work.
Conclusion
Line 573 The results clearly indicate adaptive metabolic and physiological responses among these different breeds under Mediterranean environmental conditions. HOW??
Line 574 Results showed that, despite the pasture quality and environmental climatic conditions, Mediterranean breeds tend to recover earlier than Saanen from the negative energy balance, which is consistent with their lower milk yield How do your results show this – you have not incorporated climatic conditions into your statistical model
Author Response
Dear Editor and Reviewers, first of all we need to express our gratitude for the effort in suggesting and evaluating the paper, clearly aiming to improve the quality of our draft.
As you will see, we accepted overall all the suggestions and opinions, except for few ones, where the issue was mainly a lack of clarity of the paper more than an error. So, in this case we improved clarity, trying to avoid the misunderstanding that generated doubts and suggestions by reviewers (and that potentially could generate doubts and misunderstanding by future readers).
Below we pasted the report of both the reviewers, reporting analytically in red our rebuttal or answers.
Title needs more clarity with regards to ‘genotypes clusters’
Au: We modified this term in the title and along the whole paper, also in accordance with the suggestion from reviewer 1
Mineral not mentioned as a result in the Abstract or in key search terms. Key terms need a complete re-think & goat not mentioned
Au: We integrated abstract with some results on mineral profile and revised keywords
Abstract purely descriptive with no key statistics
Au: We integrated statistical significance information
Parameters of interest mentioned but not expanded upon in terms of why they were selected for investigation in the introduction. The introduction as it stands is weak. Also lacking on information under standardised conditions on the digestive efficiency and feed conversion efficiency of the breeds of goat selected for the study. Suggested type of missing reference….(see N. Salah, D. Sauvant and H. Archimède (2014) Nutritional requirements of sheep, goats and cattle in warm climates: a meta-analysis. Animal, 8:9, pp 1439–1447 doi:10.1017/S1751731114001153)
Au: The introduction has been improved indicating the parameters selected for the investigation, and what are they evaluating.
Why were these particular breeds selected for the study – an indication of their relative influence/importance in the caprine dairy industry in the EU would be useful.
Au: The introduction has been revised to make it clear the relevance of the selected breeds. Those breeds are within the most important breeds reared in South Italy. As it was already stated in the first submission, it is important to preserve these breeds to avoid losing animal genetic resources.
Not enough emphasis on the influence of dry conditions on parameters of interest in the Introduction and the fact that the literature for plasma and milk parameters of nutritional status health production ais biased towards temperate breeds and environmental conditions. More emphasis on potential effects of climate change – could this term go in the title?
Why measure environmental conditions and not incorporate them in your statistical model? Either as raw data or as some recognised index of thermal load? (see Ingram and Mount 2012 Man and Animals in hot environments)
Au: We discussed a lot on this point and together concluded that this suggestion, although interesting and representing a further opportunity for other studies, is not possible for the present paper, for several reasons that we try to list analytically below:
- THI values are nested to DIM, because kidding was synchronized. So, stage of lactation and climatic conditions are overlapped, and it is impossible to discriminate the effect of lactation from the effect of seasonal/environmental conditions. The synchronized kidding techniques is very spread in several European countries, both to produce kids for meat production, and for allowing to animals access to the best pasture quality during the seasons of higher nutritional requirements. For a better clarity of this aspect we modified statistical model highlighting that lactation stage effect is overlapped to environmental seasonal conditions.
- The paper aims to deeply describe environmental conditions, both for allowing to reader to verify that they are consistent with Mediterranean climatic conditions (but similar to several other climatic areas, as subtropical ones), but it doesn’t aim to evaluate the effect of heat load on animal adaptative capacity. In this case the study should be differently designed: a larger number of animals, not synchronized for kidding period, and the evaluation of production and physiological patterns according to different THI classes. Besides in that case we should evaluate which is the best fitting bioclimatic index (which THI? Maximum? Minimum? Daily Average? Average of the daytime hours?) and when heat load affects physiological and production patterns (bioclimatic index of the sampling day or of some days before?). Finally, the indication of THI and nutritional value of the pasture are not been reported with the aim of describing them as "effect", but as a complete and analytic description of the environmental conditions, that were the same for all the animals. So, evaluating environmental effects as independent variables is a different paper.
- For a better clarity and aiming to reduce the amount of data of the paper, we moved environmental data (both climatic and of the pasture) in a supplementary file
Are all your parameters normally distributed? Some of the values for coefficient of variation are very high – did you check for normality in your data? Would a generalized linear model that can cope with mixed distributions of data: data with a bimodal or with a Poisson distribution be better?
Au: Thanks for this observation, as we did not highlight that the unique item not normally distributed was ALP. Particularly, residuals, post MIXED proc application were not normally distributed. So, we transformed the values of this parameter in Log10, and we verified that this transformation made these residuals normally distributed. In tables we reported LS Means of the back-transformed Log value. All these passages are clarified both in the M&M section (Statistical analysis) and as footnotes in all the tables.
Because the unique parameter that showed residuals not normally distributed was ALP we decided to run the proc MIXED, and to normalize ALP through Log10, for a better homogeneous presentation of results.
Line 48 Plasma chemistry profiles… what about parallel milk chemistry profiles . A vague sentence. There needs to be more clarification on why selected parameters were investigated. The statements about the usefulness of these plasma chemistry profiles are unreferenced.
Au: We better specified and added references
Line 82 I don’t understand the sampling procedure for nutritional content of the pasture. This would have varied considerably over the 30 week lactation cycle and yet the 9 sampling events were pooled to give one mean and variation value for each pasture variable over the course of the study. Why were pasture sampling events not tied to blood and milk sampling events? The data in Table 1 need to be broken down to each of the 9 sampling events
Au: For a better characterization of pasture quality we, following the suggestion, modified the table depicting pasture quality in winter, Spring, Summer, Autumn, better describing seasonal variability. Anyway, as already discussed for climatic conditions, we are not able to discriminate climate effect by pasture effect, nor by lactation stage, as these parameters are overlapped.
Lines 98-104 would be better tabulated
Au: Done
Table 1 and Fig 1 belong in the results section
Au: As both table 1 and Fig 1 aren’t results, but descriptive data regarding environmental conditions, we moved them in a Supplementary file.
Re-phrase lines 143-149
Au: Done
Do the addresses of the manufacturers of equipment belong in the text?
Au: Done
Lines 150-167 would be clearer tabulated
Au: Done
Lines 224-229 you should comment on lactation stage and lactation stage x genotype effects
Au: Done
Lines 321-242 Table 3 commentary. What strikes me is the lack of breed effects in the data? Would an alternative mode of statistical testing be worthwhile as you have been working with parameters with a high degree if inherent variation and possibly not a normal distribution – and have applied very stringent statistical testing for normal data. The other way forward if your data are not normal is to try transforming it but I have rarely had much luck with this approach. The most appropriate statistical test would be a better approach
Au: I understand, again, your perplexity about. As we say before, we had only ALP as not normally distributed parameter. Now it is highlighted in the text. But, we understand, there is no difference between breeds in a lot of parameters.
Lines 251-278 – this is a blow by blow account of each result in Table 3. A briefer more condensed analysis of the main findings would be more appropriate.
Au: Thank you for your suggestion. We know that the results section is particularly long because of the presence of a lot of parameters with some significative differences. We agree with your suggestion to be more condensed and to shorten some results. We revised this part and some “minor” results were deleted to condense this section and make it easier to be read. The same was done in other parts after in the text, as you suggest.
Lines 274-278 You have achieved this with your description of findings from Table 4.
Au: thank you.
Table 6, Table 7 & Table 8 Superscript the letters indicating level of statistical significance
Au: Done
Lines 329-351 again too long winded – see previous comment about describing main findings from a Table of results
Au: Also, this part was revised, as the previous part, and some minor results were deleted.
Discussion
Start discussion with the experimental aims
Au: Done
Lines 388 – 431 of Discussion; ANOVA and Descriptive Statistics; generally a descriptive comparison with data from the literature with insufficient inference – the following section is similarly unfocussed. Start from the perspective of the main themes in the experimental design: Mediterranean conditions; breed; lactation stage; (thermal load from the environment). Much of the data in literature you are making comparisons with has not been collected under comparable conditions and this is an important point to highlight within the framework of climate change.
Au: We want to highlight when the results are in a “physiological range” for the species. However, as we said before, there was not the will to investigate how climatic conditions can affect these parameters, but only to explain that climatic conditions were the same for all the groups, considering that they were in the same farm. We understand that this part is a little bit long and sometimes, to compare some parameters, we take in account also some researches conducted in lands that are characterized by different climatic conditions if compared to ours. So, we revised this part, we shortened it and we focus it on our real aim, different breeds group according milk yield and different lactation stages.
Line 394 What environmental conditions were these data gathered under? This comment applied to any citation referencing clinical data.
Au: This part was deleted.
Lines 404-406 Na and K levels can both be influenced by other factors such as hydration status; thermal load? So I don’t understand this statement ?
Au: Yes, we agree with you about the possibility that these two parameters can be affected by thermal load and obviously by hydration status. However, both these two conditions are equal in our experimental breeds groups, so we try to observe if the group and the lactation stage are able, in the same environmental conditions, to affect these and more other parameters. This sentence explains how, for our findings, no differences due to these extrinsic factors (stage of lactation), can be reported. We modified the sentence to better fit this concept.
General comment for Discussion: You are not focussing on the environmental impact on the parmeters of interest. . This could also be incorporated as an index of thermal load into your statistical model. Environmental impact (or influence of thermal load) on the data is a novel and relevant theme that has been generally neglected in the work.
Au: Thank you very much for your suggestion. We appreciate it and we agree with you about the importance of the influence of thermal load on animal welfare and production. However, as you can see before, we try to better explain that it was not the aim of this paper. In order to study the thermal load, we need more and more lactations and test day controls for milk yield and quality and for blood parameters. These data are not enough for that kind of study. Moreover, these animals were bred all together, in the same farm, with the same thermal conditions, with the same seasonality of parity. There are three breed groups (according milk yield) and this sure negatively affect a study on thermal tolerance and/or thermal load effect (it must be considered one breed and thousands of checks).
Conclusion
Line 573 The results clearly indicate adaptive metabolic and physiological responses among these different breeds under Mediterranean environmental conditions. HOW??
Au: Thank you for this, we agree. The sentence was deleted.
Line 574 Results showed that, despite the pasture quality and environmental climatic conditions, Mediterranean breeds tend to recover earlier than Saanen from the negative energy balance, which is consistent with their lower milk yield How do your results show this – you have not incorporated climatic conditions into your statistical model
Au: As we say before, the thermal condition was the same for all the animals. All animals were in the same farm and had the same feeding conditions. So, in the conclusion, we underline that, although are breeds with a different yield potential, they can adapt differently at the same thermal condition. The autochthonous breeds probably faster than not autochthonous ones. However, the conclusion section was partially revised.